# Tandem Repeat DNA Provides Many Cytological Markers for Hybrid Zone Analysis in Two Subspecies of the Grasshopper *Chorthippus parallelus*

**DOI:** 10.3390/genes14020397

**Published:** 2023-02-03

**Authors:** Beatriz Navarro-Domínguez, Josefa Cabrero, María Dolores López-León, Francisco J. Ruiz-Ruano, Miguel Pita, José L. Bella, Juan Pedro M. Camacho

**Affiliations:** 1Departamento de Genética, Facultad de Ciencias, Universidad de Granada, 18071 Granada, Spain; 2Department of Organismal Biology—Systematic Biology, Evolutionary Biology Centre, Uppsala University, SE-752 36 Uppsala, Sweden; 3School of Biological Sciences, University of East Anglia, Norwich Research Park, Norwich NR4 7TU, UK; 4Departamento de Biología (Genética), Facultad de Ciencias, Universidad Autónoma de Madrid, 28049 Madrid, Spain; 5Centro de Investigación en Biodiversidad y Cambio Global (CIBC-UAM), Universidad Autónoma de Madrid, 28049 Madrid, Spain

**Keywords:** *Chorthippus parallelus*, hybrid zone, repetitive DNA, tandem repeats, FISH

## Abstract

Recent advances in next generation sequencing (NGS) have greatly increased our understanding of non-coding tandem repeat (TR) DNA. Here we show how TR DNA can be useful for the study of hybrid zones (HZ), as it serves as a marker to identify introgression in areas where two biological entities come in contact. We used Illumina libraries to analyse two subspecies of the grasshopper *Chorthippus parallelus*, which currently form a HZ in the Pyrenees. We retrieved a total of 152 TR sequences, and used fluorescent in situ hybridization (FISH) to map 77 families in purebred individuals from both subspecies. Our analysis revealed 50 TR families that could serve as markers for analysis of this HZ, using FISH. Differential TR bands were unevenly distributed between chromosomes and subspecies. Some of these TR families yielded FISH bands in only one of the subspecies, suggesting the amplification of these TR families after the geographic separation of the subspecies in the Pleistocene. Our cytological analysis of two TR markers along a transect of the Pyrenean hybrid zone showed asymmetrical introgression of one subspecies into the other, consistent with previous findings using other markers. These results demonstrate the reliability of TR-band markers for hybrid zone studies.

## 1. Introduction

Eukaryotic genomes contain a significant fraction of repetitive DNA, which can be distributed in interspersed or tandem patterns. Interspersed repeats are typically transposable elements (TEs), whereas tandem repeats (TRs) may be coding (e.g., gene families such as histone or ribosomal genes) or non-coding (satellite DNA). The Library Hypothesis explains the conservation of TR sequence over long evolutionary periods [1]. Its three main proposals are: (i) some members of the library can be amplified in a given species but remain at low levels in other species, (ii) most evolutionary changes in satDNA are quantitative, and iii) the appearance of new satDNAs would represent the amplification of pre-existing satDNA in the library, rather than de novo appearance previously proposed by [2]. Next-generation sequencing (NGS) analysis in the grasshopper *Locusta migratoria* identified numerous non-coding TR families within a single genome [3]. Interestingly, *L. migratoria* and *Oedaleus decorus*, which share a most recent common ancestor about 23 Mya, do not share around two thirds of their TR catalogue [4], suggesting the possibility of de novo origin for tandem repeats as proposed by [2]. However, satDNA library analyses at shorter time scales are scarce. For example, [5] analysed three satDNA families previously described in *Schistocerca gregaria* by [6] in nine other *Schistocerca* species that diversified from *S. gregaria* during the last 7.9 Myr across the American continent. This analysis, while limited to just three families, demonstrated the role of chance events in satDNA evolution and that changes are mostly quantitative, closely fitting the Library Hypothesis predictions.

Two subspecies of the grasshopper, *Chorthippus parallelus, C. p. parallelus* (CPP) and *C. p. erythropus* (CPE), form a hybrid zone (HZ) in the Pyrenees. This HZ has been studied using various approaches, including nuclear and mitochondrial genes, behaviour, morphology, *Wolbachia* infection, and transcriptomics. Of particular relevance for our present study is the observation that some differences exist between the two subspecies in terms of the distribution and composition of heterochromatin on their chromosomes [7,8,9]. The relationship between heterochromatin and non-coding tandem repeats is well established. In some cases, TRs might be necessary for the proper assembly of constitutive heterochromatin surrounding centromeres and telomeres, without which cell division can be disrupted [10]. In other cases, some satDNAs are transcribed into non-coding RNAs that may have unknown but important functions [10,11]. In the case of CPP and CPE, the differences in the distribution and composition of the heterochromatin are particularly noteworthy, because they concern the X-chromosome involved in the X0♂/XX♀ sex determination system. The X chromosome shows a distal heterochromatic band in CPP that is absent in CPE. However, in CPE, the X-chromosome displays an exclusive interstitial band that cannot be explained by an inversion, but rather by the recent amplification and heterochromatinization of specific sequences [7,12].

The library hypothesis was proposed by Fry and Salser for satellite DNA [1], but is applicable to any type of non-coding tandem repeat (TR). It suggests that species within a given taxonomic group (e.g., acridid grasshoppers) share a common library of satellite DNA sequences (or any other TR) whose evolutionary changes are mostly quantitative. On this basis, it predicts that TRs should show low divergence between individuals, populations, or subspecies at the intraspecific level. However, not many genomes have been thoroughly examined at these levels. In order to contribute to fill this gap, we analyse here a large set of non-coding TRs in the CPP and CPE subspecies of the grasshopper *C. parallelus*, from two purebred natural populations located on the opposite sides of a well-studied hybrid zone (HZ) in the Pyrenees [7,8,13,14,15,16,17,18,19,20,21]. It is important to note that some TEs contain TRs, and many satellite DNA families show sequence homology to TEs (see [22] for review). On this basis, we will refer here to TRs in general, whether they are satellite DNA or part of TEs, as both can provide cytological markers that are the main purpose of the present research.

Cline analysis [23,24] is a common approach used to study HZs, with the goal of identifying the genes involved in species boundaries. In the particular case of the Pyrenean HZ formed by CPP and CPE, cline analysis has provided important insights into the dynamics of some quantifiable traits, such as morphological, mitochondrial, cytogenetical, or behavioural characteristics. However, the clines for different traits are not always concordant or coincident, or aligned with the geographical centre of the HZ. It is important, then, to uncover processes such as introgression, selection for or against certain traits, the existence of barriers to gene flow, gene exchange events, etc. [8,13,14,21,25,26]. Therefore, any additional markers suitable for cline analyses can provide valuable additional information to compare with previous clines, particularly if they are related to the heterochromatin content of the chromosome complement, which has been found to differ between these subspecies [7,8,9]. With this in mind, the main objective of this research was to develop new cytological markers for HZ analysis and to preliminarily test their utility by using two of them for the characterization of a geographical transect where the two subspecies come into contact. 

## 2. Materials and Methods

Specimens of *C. parallelus* were collected during the summer of 2016 at seven natural populations along the Col de Portalet transect in one of the contact zones of the Pyrenean hybrid zone [7], including pure populations at Arudy (France) and Escarrilla (Spain) (Table 1). The grasshoppers were collected with an entomological sleeve, being immediately sacrificed. Testes were fixed in 3:1 ethanol:acetic acid and body remains were kept in pure ethanol at −20 °C until used. To search for TR sequences, we used two low-coverage libraries previously reported in [20], which had been generated from a female from each subspecies collected at Arudy (CPP; NCBI-SRA accession no. SRX9529342) and Escarrilla (CPE; NCBI-SRA accession no. SRX9529340). 

We ran the satMiner pipeline [3] to identify TRs in the CPP and CPE genomes. Prior, we performed a quality trimming with Trimmomatic [27], using the options “ILLUMINACLIP:TruSeq3-PE.fa:2:30:10 LEADING:3 TRAILING:3 SLIDINGWINDOW:4:20 MINLEN:101”. Then we ran eight rounds of satMiner pipeline, which implies clustering with RepeatExplorer v0.9.7.8 (http://repeatexplorer.umbr.cas.cz/, accessed on 7 December 2016) and filtering with DeconSeq v0.4.3 (https://deconseq.sourceforge.net/, accessed on 5 May 2013), starting with 100,000 randomly selected read pairs and duplicating the number of reads after each round. Then, we joined all the resulting TR sequences and generated dimers, or a higher number of repeats when necessary to reach at least 200 nt, using the dimerator.py script (https://github.com/fjruizruano/ngs-protocols/blob/master/dimerator.py, accessed on 29 June 2018). Next, we searched for homology between sequences with the rm_homology.py script (https://github.com/fjruizruano/satminer/blob/master/rm_homology.py, accessed on 2 October 2015) to manually select a consensus for sequences with an identity higher than 95%. We considered these consensus sequences as subfamilies (or variants) belonging to a same family if they displayed 80% or higher identity. Finally, TR families showing some sequence homology but showing less than 80% identity were considered as members of the same superfamily [3].

We used the resulting collection of TRs as reference to estimate abundance and divergence with the RepeatMasker software [28] using 5 million read pairs that we randomly selected with seqtk (https://github.com/lh3/seqtk, accessed on 1 June 2015) in each library. We numbered the TR families in order of decreasing abundance in the CPP library. These sequences are deposited in GenBank with accession numbers OQ129773—OQ129924. We analysed possible homology of the consensus sequences, obtained for the different TR families, with other repeats from Orthoptera by using RepeatMasker searches in RepBase 27.04 (last accessed 27 April 2022) [29].

Repeat landscapes typically represent abundance of sequences in 1%-identity intervals. However, for higher accuracy, we estimated TR abundance in 0.5%-identity intervals using the “*.align” files from RepeatMasker. For this purpose, we wrote a custom script (https://github.com/fjruizruano/ngs-protocols/blob/master/repeat_landscape_decimal_050.py, accessed on 28 October 2019). We finally analysed the repeat landscapes to estimate the relative peak size (RPS) and divergence value showing the maximum abundance in a repeat landscape (DIVPEAK) using the SatIntExt pipeline (https://github.com/fjruizruano/SatIntExt, accessed on 7 April 2021 see [4] for details).

To generate probes for FISH analysis, we designed divergent PCR primers to amplify 82 TR families using Primer3 [30], with Tm ~60 °C (Appendix A); 77 of them yielded PCR products, which were then labelled with fluorochromes as described in [3,31]. The slides were visualized in an Olympus BX41 microscope equipped with a DP70 digital camera. FISH mapping was performed on spermatocytes of a total of 28 males from Arudy (CPP) and 28 males from Escarrilla (CPE). Finally, to study introgression, we performed FISH analysis on specimens collected at seven locations along the Col de Portalet transect [7]. For this purpose, we used two non-coding TR markers: CpaTR100-295 (autosomal, 71 males), and CpaTR104-269 (X-linked, 61 males). 

For statistical analysis, matched-pair comparisons were performed by the Gardner–Altman estimation plot method devised by [32] following the design in [33], as implemented in https://www.estimationstats.com (accessed on 15 December 2022). This analysis calculates the effect size by the mean difference between groups for paired mean differences. The effect size is then evaluated by the 95% confidence interval (95% CI) and whether it includes or not the zero value.

## 3. Results

### 3.1. Searching for Tandem Repeats Displaying FISH Bands

After 8 rounds of the satMiner protocol, we found 152 different subfamilies (i.e., variants) of tandem repeat elements in the two Illumina libraries analysed from the CPP and CPE subspecies, which were grouped into 110 families and 13 superfamilies (Appendix A). The telomeric repeat was the 19th family in order of decreasing abundance in CPP, and it showed the typical repeat unit length (RUL) of five nucleotides found in other grasshoppers [3]. For subsequent analyses, the telomeric repeat was excluded. RUL displayed by the remaining 109 TR families ranged between 6 and 405 bp (Appendix A). The CpaTR016-7 and CpaTR105-6 families were actually microsatellites with RULs equal to 7 and 6 bp, respectively, the latter apparently showing arrays large enough to display a FISH banded pattern (see below and Appendix A). Both families could be appropriate to develop microsatellite markers for the analysis of molecular polymorphisms, in addition to the nine microsatellite sequences reported by [34]. Remarkably, when the consensus sequences of the 110 TR families were annotated against a collection of repeats from Orthoptera retrieved from Repbase 27.04 [29], 21 of them (19%) showed sequence homology with transposable elements (TEs) (see Appendix A). CpaTR036-168 showed homology with the PST1 satellite previously described in *Stauroderus scalaris* with the same RUL [35]. 

A total of 77 families showed successful PCR amplification, 50 of which yielded FISH bands i.e., banded (B) or dotted-banded (DB) patterns, on one or more chromosomes in at least one of the subspecies (Figure 1 and Table 2 and Table 3). The remaining 27 showed dotted (D) or no signal (NS) patterns (see [36] for further description of FISH patterns). In CPP, 39 families yielded FISH bands: 38B + 1DB. In CPE, however, 46 families showed FISH bands: 45B + 1DB. TR spread across chromosomes varied very much among TR families, with most of them being located on a single chromosome pair (20 in CPP and 27 in CPE) and only two (CPP) or three (CPE) reaching the nine chromosome pairs (Figure 2). 

### 3.2. Differences between Subspecies

Among several molecular parameters analysed (abundance, divergence, TSI, RPS, and DIVPEAK; see Methods and [4]) (Appendix A), the 50 families showing FISH bands in one or both subspecies displayed lower sequence divergence and higher tandem structure in CPE compared to CPP (paired mean difference for divergence: −1.03 (95% CI: −2.6, −0.218); paired mean difference for TSI: 0.0563 (95% CI: 0.0206, 0.13)). The same test did not reveal global differences between subspecies for the three other parameters (not shown). Interestingly, the differences between subspecies were concentrated into 15 out of the 50 TR families, as 35 of them showed the same FISH pattern in both subspecies (see Table 2 and Table 3). The remaining 15 families showed the B pattern in one subspecies and the NS one in the other (4 TRs in CPP and 11 TRs in CPE), and thus provided the most interesting cytological markers for HZ analysis. In CPE, the extra TR-bands were present on a single chromosome pair for ten TR families (CpaTR033-222, CpaTR034-61, CpaTR050-288, CpaTR057-102, CpaTR076-168, CpaTR094-170, CpaTR097-108, CpaTR100-295, CpaTR103-33 and CpaTR107-237) and on three chromosome pairs for CpaTR047-287 (Table 2 and Table 3) presumably due to spread between non-homologous pairs. Likewise, three (CpaTR013-55, CpaTR053-405 and CpaTR058-196) out of the four TR families showing FISH bands in CPP which are absent in CPE were found on a single chromosome pair, whereas the remaining TR (CpaTR025-248) was present on two chromosome pairs (see Table 2 and Table 3). The scarce spread to other non-homologous chromosomes shown by 13 of these 15 differentially amplified TRs suggests that the corresponding amplification events occurred recently, presumably during Pleistocene glaciations which provided the geographical isolation needed for the differentiation of these two subspecies. 

The X chromosome in the CPP subspecies showed a distal heterochromatic band which was absent from the X in CPE [7,12] and contains ribosomal DNA in CPP only [37]. Table 2 and Table 3 show that the distal region of the X chromosome also harbours, only in CPP, two TR families yielding large FISH bands, specifically CpaTR017-289 and CpaTR032-20 (Figure 3).

The differences observed between subspecies were unevenly distributed within the genome, as autosomes 1 and 7 showed more TR families in CPE (5 and 7, respectively) than in CPP whereas autosomes 6 and 8 showed more TR families in CPP (4 each) (Figure 4). 

### 3.3. Correspondence between Molecular and Cytological Parameters

The graphical representation of genomic abundance against divergence from a reference is known as repeat landscape (RL). It allows the identification of amplification events at the molecular level, which appear as a peak of abundance showing the same divergence with respect to the consensus sequence. This divergence value (i.e., DIVPEAK) reflects the time since the last TR amplification (see [4]). Whereas leptokurtic RLs indicate recent amplification, platykurtic ones indicate that TR sequences are rather degenerated due to absence of recent amplification events. The comparison of RLs between TR families displaying the NS pattern in one subspecies and the B pattern in the other one revealed the simultaneous presence of amplification peaks and FISH bands in the subspecies displaying the B-FISH pattern (see two examples in Figure 5). In fact, the 11 TR families showing the B pattern in CPE (and NS in CPP) displayed higher homogenization and lower degeneration indexes than their orthologous TRs in CPP (Figure 6), providing a nice example of how amplification leads to higher homogenization of TR sequences and it delays degeneration.

### 3.4. Analysis of the CPP-CPE Contact Zone Using two TR Markers

To test the utility of the TR markers for the study of hybrid zones, we chose two families showing differential presence of FISH bands between CPP and CPE. The first one, CpaTR100-295, showed the B pattern in CPE, with a single band on autosome 7, and the NS one in CPP (Figure 7a). The other family, CpaTR104-269, showed only a minute band on autosome 6 in CPP, but it showed bands on four chromosomes in CPE, with minute bands on autosomes 2 and 7 and large bands on autosome 6 and the X chromosome (Figure 7b). For the present analysis, we only used the large interstitial band on the X chromosome, which was completely absent in CPP, as a sex-linked marker.

In addition to the two pure populations at either end of the contact zone (Arudy and Escarrilla), i.e., those where the Illumina sequencing and the previous FISH analysis were conducted, we analysed five intermediate sites along a transect connecting these locations (Figure 8a). The frequency of banded (B) and non-banded (N) chromosomes observed at the seven sites analysed (Table 4 and Figure 8b) was consistent between the CpaTR100-295 and CpaTR104-269 markers (paired mean difference for the frequency of banded chromosomes: −0.0473 (95% CI: −0.131, 0.0124)). The highest altitude on the transect corresponds to the POR population, which is the geographical centre of the hybrid zone, as it is the point of longer separation by ice between subspecies during the last glaciation [38,39]. Examples of introgression for the CpaTR100-295 marker are shown in Figure 8c,d. Remarkably, the two TR markers showed an inflection point in frequency changes of banded chromosomes along the transect at the CM population (see Figure 8b). This suggests asymmetrical gene flow, with higher introgression from CPP toward CPE than the reverse. This is also supported by other markers in this HZ which have shown an abrupt change in their clines at this population [8,13,14,16]; see also [21]. The CM population therefore appears to be a key point in the hybrid zone.

## 4. Discussion

A surprising result in our research was the observation that about 19% of the 110 TR families identified in *C. parallelus* showed sequence homology to known TEs from Orthoptera in RepBase. According to recent reports, there might be a structural relationship between these two kinds of interspersed and clustered repetitive DNA [22]. For example, [40] found three satDNA sequences in the Asian swamp eel (*Monopterus albus*) showing partial homology with TEs, and [41] made a similar finding for a satDNA family in plants genus *Chenopodium*. In *C. parallelus*, the high number of TR families with sequence homology to TEs suggests that some of the 110 families may actually be TR-carrying TEs, which could explain why they did not yield FISH bands, as the interspersed distribution of the TEs conditions an interspersed distribution also for their TRs, even though the latter is clustered within the TE. Further research is needed to fully understand the relationship between TRs and TEs in *C. parallelus*, as this species appears to be an ideal material for investigating the potential contribution of mobilome to satellitome.

Our present results support the recent demonstration that a single satDNA can display the B or NS patterns at both intra- and interspecific levels [4]. The additional TR-bands found in only one subspecies (11 TRs in CPE and 4 in CPP) are likely the result of local amplification. The fact that the 11 TR families which showed recent amplification in CPE and yielded FISH bands (which are absent in CPP) displayed higher homogenization and lower degeneration indices provides new evidence for the hypothesis that each new satDNA amplification burst leads to its rejuvenation by preventing it from degenerating and disappearing [4].

The amplification of non-coding tandem repeats involves the large-scale duplication of repeats, which typically occurs through unequal crossover [2,42]. As shown by [4], amplification is a strong homogenizing force, since the new copies are essentially identical at the beginning and accumulate sequence differences over time by point mutations. The authors of [43] established that a minimum target DNA of about 1 kb is necessary for FISH visualization. On this basis, we can infer that TR families displaying the B-FISH pattern in only one subspecies underwent molecular clustering to form arrays surpassing this minimum threshold during their allopatric isolation on either side of the Pyrenean barrier, making them suitable markers for hybrid zone analysis.

The TR families displaying B or NS patterns on one subspecies or the other were actually present in both Illumina libraries, as indicated by abundance estimations (see Table 3). Therefore, they formed FISH bands only in one subspecies due to local amplification during the separate evolution of both subspecies. As these *C. parallelus* subspecies diverged recently (1 Mya; see [44]), it is reasonable to date the formation of these FISH bands through massive local amplification during this time frame. The higher number of unique haplotypes for a noncoding nuclear DNA marker sequence in Turkey compared with the Balkans and Europe led [17] to conclude that *C. parallelus* populations in Turkey evolved in effective allopatry for a long time. On this same basis, we speculate that the higher number of TR amplifications in the Iberian endemism CPE may indicate longer isolation of Iberian populations during Pleistocene [44].

The analysis of the hybrid zone using two of the TR markers developed in this study showed consistency between both markers, with the cline being centred in the first Spanish population analysed (CM). This population (CM) is beyond the geographical centre (POR), which is the highest site, and also the altitudinal limit for the survival of this organism. In fact, the POR population is located at the transition point between the French and Spanish valleys that meet in this mountain pass, forming the corridor where the HZ was formed. Thus, the observed cline indicates asymmetrical introgression, with CPP markers reaching high altitude localities such as POR (French side) and CM (Spanish side) (1708 m and 1569 m, respectively). Higher altitudes are expected to have shorter snow-free seasons, due to colder temperatures. This is very relevant for the life cycle of these grasshoppers: prior to the first snowfall, the adults need to lay their eggs in the soil, which will enter diapause until the snow melts in the next spring. 

The cline observed here from the TR markers is consistent with the cline for hybrid sperm dysfunction found by [16], which is wider in the Spanish side (6.2 versus 4.2 Km). Other traits showing a cline displaced towards the South include male wing ratio, peg density, and female pronotum ratio [45]. Similar introgression has also been observed for two markers: mtDNA in Escarrilla [46], and for an X chromosome marker characteristic of CPP individuals, which was found in Iberian populations about 400 km to the NW of the Pyrenean hybrid zone [9]. This and other evidence support the hypothesis that gene flow is at least partially responsible for the introgression after recent secondary contact between these two ancestral taxa that diverged in allopatry [21].

The Fast-X effect increases the role of the X chromosome in adaptation and speciation [47], and perhaps the observed differences between subspecies in the number of TRs might be related to the implication of the sex chromosome in subspecies differentiation. The most conspicuous difference is the distal heterochromatic band, observed only in CPP, which contains ribosomal DNA [37] and two TR families (CpaTR017-289 and CpaTR032-20). Unfortunately, both TR families were also present on other chromosome pairs in both subspecies, so that the comparison of molecular parameters between subspecies (see Table 3) is hindered by the other TR clusters on the autosomes. Elucidating the precise structural relationship between these two kinds of tandem repeats is an interesting prospect for further research. Interestingly, CpaTR058-196 displayed a small interstitial FISH band only on the CPP X chromosome, which was absent in CPE and recently amplified in CPP (see Table 2). This could be an interesting marker for future studies. 

Isozyme analysis of several *Chorthippus* species, along with the two subspecies of *C. parallelus* analysed here (CPP and CPE) showed that these subspecies have been diverging since they were separated by the Pyrenean ice sheet during the Pleistocene, about 1 Mya [44]. In contrast, analysis of the mitochondrial cytochrome oxidase subunit I (COI) gene across European populations, including samples from the three main glacial refuges in the Iberian, Italic and Balkan peninsulas, indicates that ancestors of the CPP and CPE subspecies remained isolated for about 476 to 506 thousand years during five or six glacial cycles [46,48], assuming mtDNA sequence evolution rate equal to 2% per million years [49]. However, more recent studies estimate that divergence between both subspecies occurred between 127,900 and 65,900 years ago, at the beginning of the last glacial cycle in Europe [21]. Our present analysis reveals that local amplification of many TR families occurred differentially between subspecies, resulting in a B pattern for TRs that previously showed the NS pattern. This occurred almost three times more frequently in CPE than CPP (11 vs. 4, respectively). These events might be useful to estimate the divergence time using the 11 orthologous TR pairs that changed from NS pattern in CPP to B pattern in CPE, based on the average turnover rate of 1.1% per million years calculated by [4] for 20 orthologous pairs of satDNA families in two oedipodine grasshoppers. The average degeneration index (DIVPEAK) for the 11 orthologous TR families in CPE was 3.3% which, when converted to time by the equation: time = DIVPEAK/(2*1.11), resulted in 1.5 Ma. When this same calculation was performed for the same 11 orthologous TR families in CPP (all showing the NS pattern), the average DIVPEAK was 7.8%, yielding a time of 3.5 Ma. This sets the differential amplifications in CPE during the Pleistocene separation of both subspecies through the recurrent glaciation periods. Interestingly, two TR families have spread between non-homologus chromosomes, while the remaining 13 are confined to a single chromosome pair, which would be consistent with amplifications of different TR families at different glaciation cycles.

Finally, as divergence between CPP and CPE occurred during Pleistocene glaciations, which featured successive cold and warm periods that led to important retractions and expansions to *C. parallelus* populations, it is possible that these demographical changes could have facilitated the fixation of TR amplifications, in resemblance with those suggested during locust outbreaks [4].

Our present analysis has shown that TR markers can be useful for HZ analysis. As the main intra- and inter-population variation for TRs is quantitative, as predicted by the library hypothesis (Fry and Salser 1977), some TRs can display differential amplification in different populations, thus providing markers in the form of FISH bands which are very useful in monitoring the evolution of diverging populations or subspecies. In the case of *C. parallelus*, further studies involving other locations in the Iberian Peninsula and the rest of Europe would provide appealing data on the evolutionary history of this interesting organism, while also offering valuable information on the possible multiple glacial refugia and perhaps the role of these sequences as reproductive barriers in the origin of the HZ or in the known infertility of the F1 hybrid males from crosses between pure subspecies [50,51].

## Figures and Tables

**Figure 1 genes-14-00397-f001:**
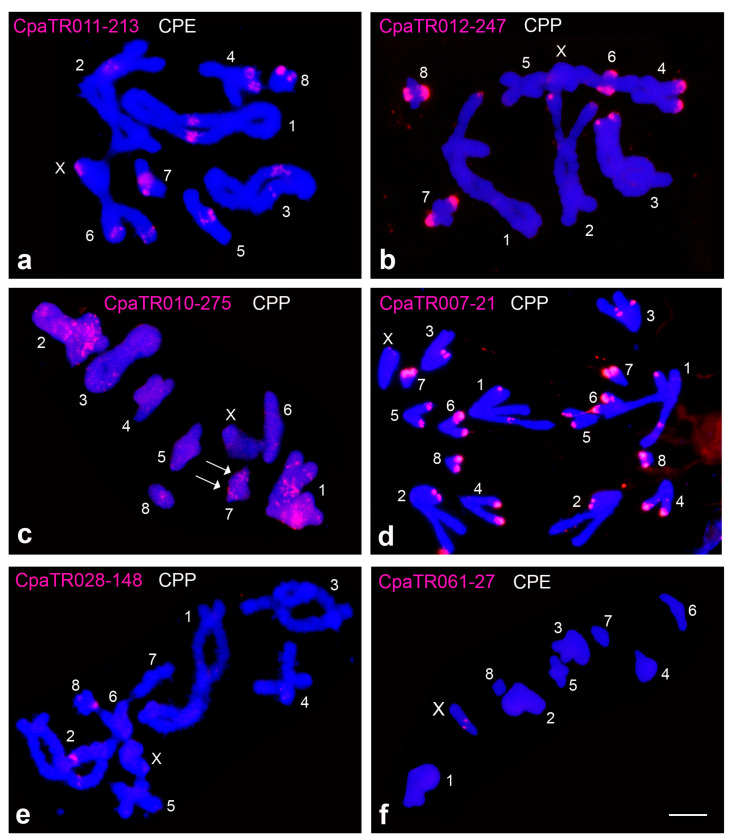
Examples of physical location of TRs by FISH in *C. parallelus*: pericentromeric bands on all chromosomes (**a**), distal bands on all chromosomes, except autosome 5 (**b**), dotted-banded (DB) pattern with a defined band on autosome 7 (arrows) (**c**), distal bands on all chromosomes (and interstitial band on autosomes 1 and 4) (**d**), pericentromeric bands on chromosomes 2 and 8 (**e**), and interstitial band only on the X chromosome (**f**) (see also Table 2 and Table 3). CPP = *Chorthippus parallelus parallelus*; CPE = *Chorthippus parallelus erythropus*. Bar = 5 µm.

**Figure 2 genes-14-00397-f002:**
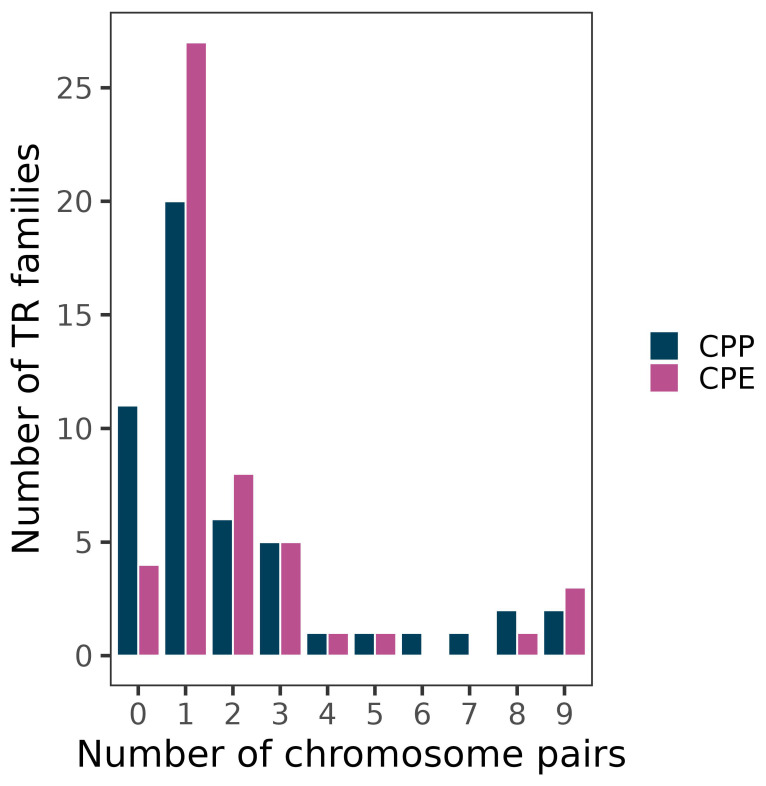
Spread of TR families among non-homologous chromosome pairs. Note the predominance of families being located on only one chromosome pair (20 in CPP and 27 in CPE).

**Figure 3 genes-14-00397-f003:**
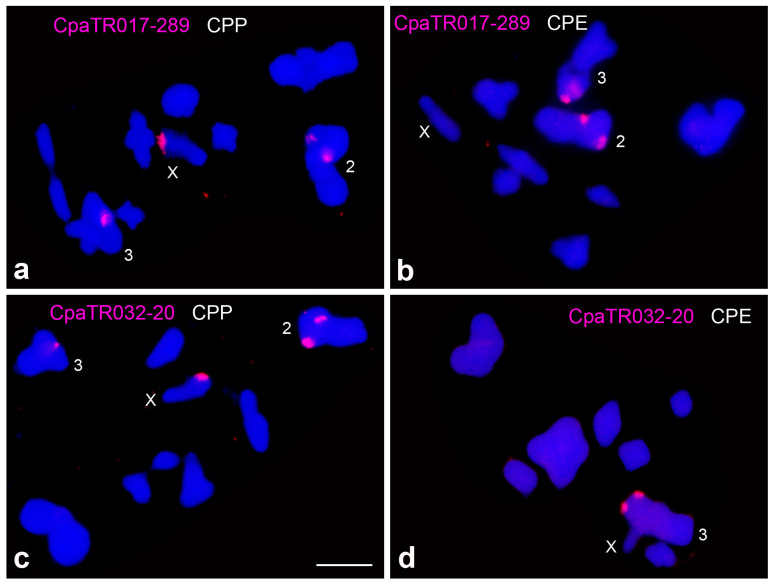
Presence of two TR families on the distal region of the X chromosome in CPP, which are absent in CPE: CpaTR017-289 (**a**,**b**) and CpaTR032-20 (**c**,**d**). Note that these TRs are also present on autosomes 2 and 3 (see also Table 2 and Table 3). Bar = 5 µm.

**Figure 4 genes-14-00397-f004:**
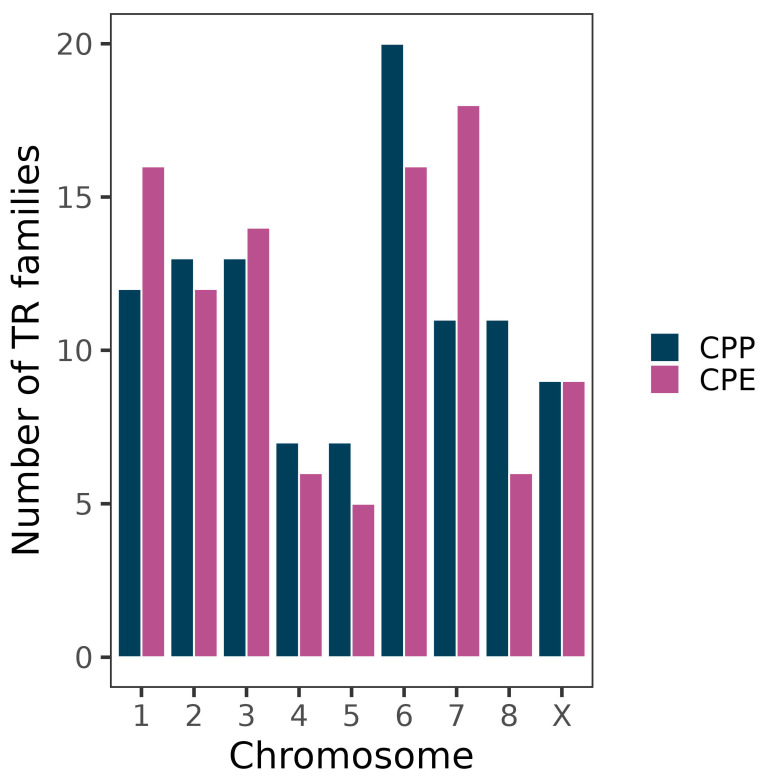
Number of TR families forming FISH bands on the chromosomes of the grasshopper *C. parallelus*. Note TR relative enrichment on autosomes 1 and 7 in CPE and autosomes 6 and 8 in CPP, compared to the other subspecies.

**Figure 5 genes-14-00397-f005:**
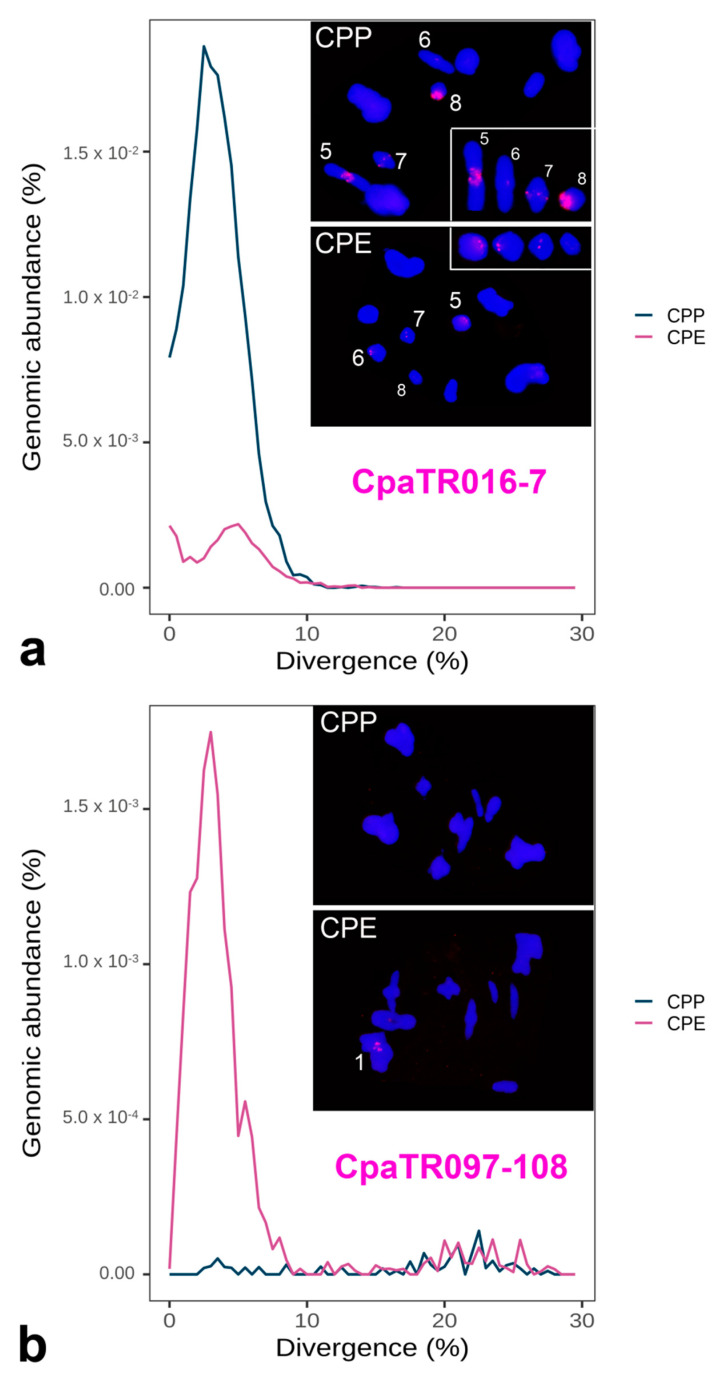
Combined repeat landscapes (RLs) and FISH patterns for two TR families showing higher amplification in one subspecies, as inferred from its higher RL peak paralleled by larger FISH bands, with CpaTR016-7 showing higher amplification in CPP (**a**) and CpaTR097-108 in CPE (**b**). The insets in (**a**) display the four chromosomes showing FISH bands for CpaTR016-7 in one or both subspecies (see also Table 2 and Table 3). Bar = 5 µm.

**Figure 6 genes-14-00397-f006:**
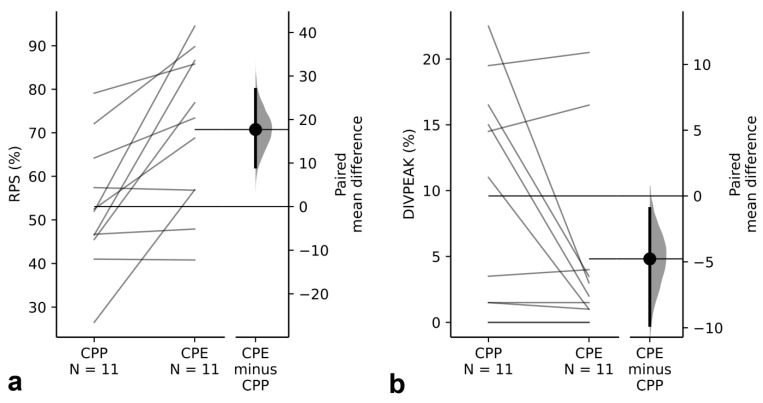
Between-subspecies paired comparison of RPS (homogenization index) and DIVPEAK (degeneration index) for the 11 orthologous pairs of TR families showing the NS pattern in CPP and B pattern in CPE. Note how these TRs show (**a**) higher homogenization (paired mean difference for RPS between CPP and CPE = 17.7 (95%CI: 9.1, 26.9)), and (**b**) lower degeneration (paired mean difference for DIVPEAK between CPP and CPE = −4.77 (95%CI: −9.82, −0.955)) for the orthologous TRs displaying the B pattern in PPE.

**Figure 7 genes-14-00397-f007:**
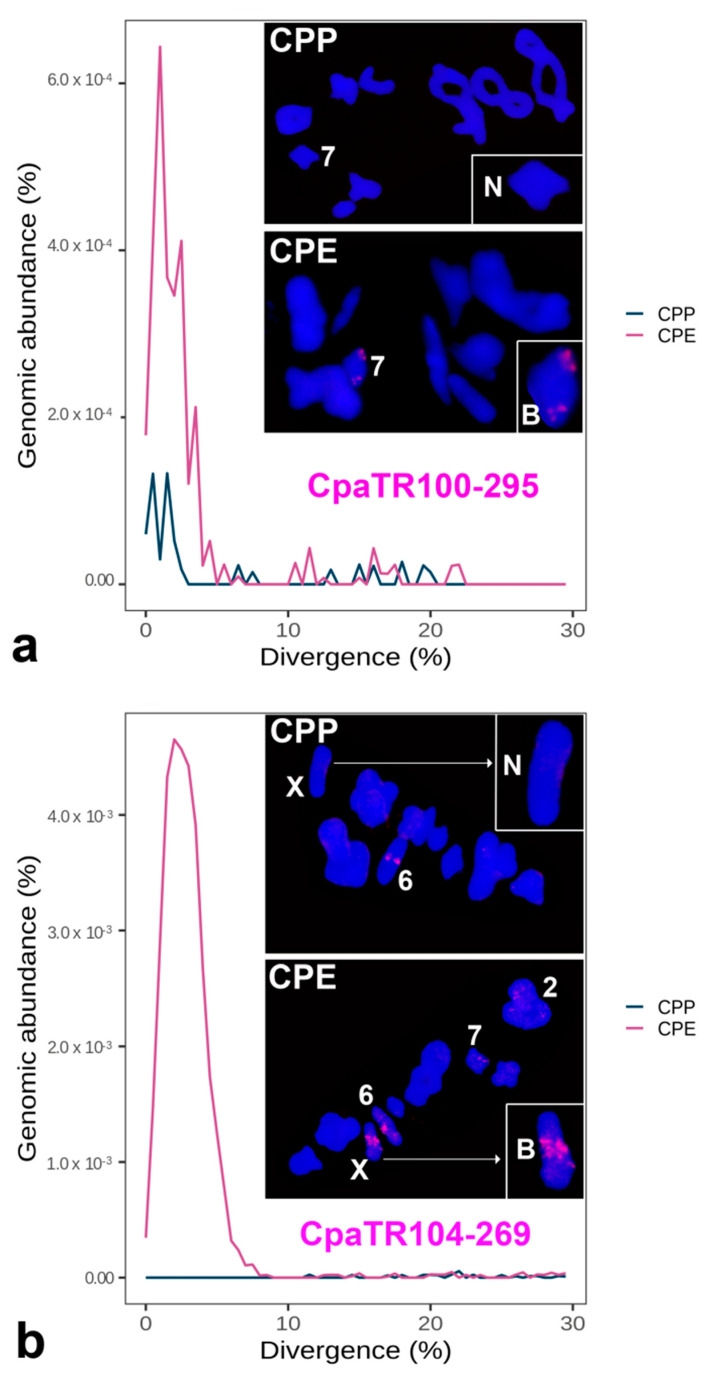
Combined repeat landscapes (RLs) and FISH patterns for the two TR families chosen for hybrid zone (HZ) analysis: CpaTR100-295 (**a**) and CpaTR104-269 (**b**). The insets show the non-banded (N) or banded (B) chromosome variants. Bar = 5 µm.

**Figure 8 genes-14-00397-f008:**
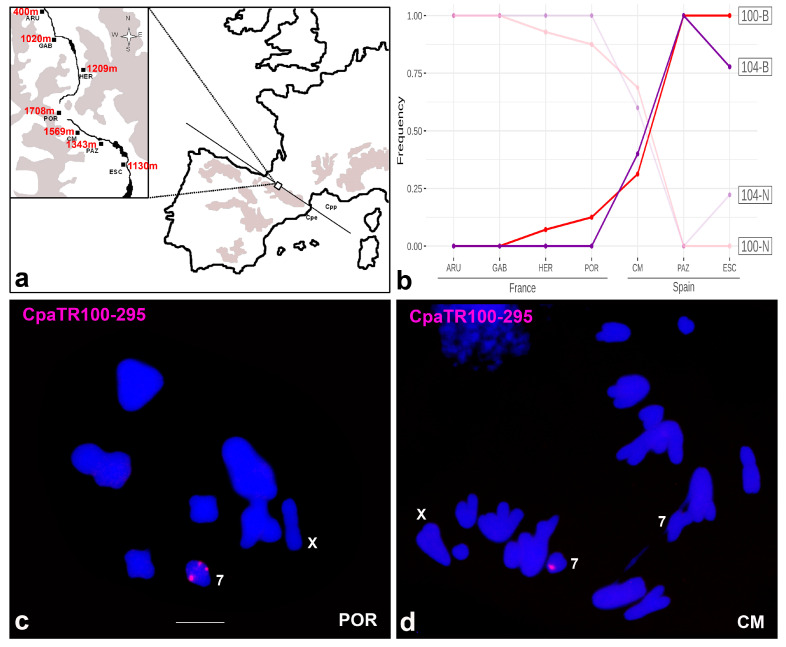
Analysis of the hybrid zone with the CpaTR100-295 and CpaTR104-269 markers. (**a**) Map showing the location of the hybrid zone at two levels of magnification. The localities analysed were ARU (Arudy), GAB (Gabas), HER (L’Hermine) and POR (Portalet) in France, and CM (Corral de Mulas), PAZ (Camino Pazino) and ESC (Escarrilla) in Spain. (**b**) Concurrent clines for both TR markers reveal that the hybrid zone is centred on the Spanish CM population, whereas the highest altitude (i.e., the geographical centre) was the POR population, on the French side. This suggests higher introgression towards the Spanish populations. (**c**) Metaphase I cell from a POR male being homozygous for the band on autosome 7 for CpaTR100-295, as is typical of the Spanish populations. This is an example of introgression of Spanish banded chromosomes into the French POR population. (**d**) Anaphase I cell from a CM male being heterozygous for the same TR, as an example of introgression of non-banded French chromosomes into the Spanish CM population. Bar = 5 µm.

**Table 1 genes-14-00397-t001:** Locations sampled at the Portalet contact zone between the CPP and CPE subspecies of the grasshopper *Chorthippus parallelus* in the Pyrenees, including two purebred samples at Arudy for CPP and Escarrilla for CPE. N = Number of males.

Country	Site	Latitude	Longitude	Altitude (m)	N
France	Arudy (ARU)	43°06′01″ N	0°26′38″ W	400	41
	Gabas (GAB)	42°53′60″ N	0°25′60″ W	1020	7
	L’Hermine (HER)	42°51′46.8″ N	0°23′30.4″ W	1209	7
	Portalet (POR)	42° 48′03″ N	0°24′54″ W	1708	12
Spain	Corral de Mulas (CM)	42°47′09.4″ N	0°23′34.4” W	1569	8
	Camino Pazino (PAZ)	42°45′57.5″ N	0°20′33.9″ W	1343	10
	Escarrilla (ESC)	42°43′54.1″ N	0°18′39.3″ W	1130	42

**Table 2 genes-14-00397-t002:** Chromosome location of 50 TR families in the CPP subspecies of *C. parallelus*. Asterisks (*) indicate that bands were observed on only one member of a chromosomal pair. Chromosome location (pattern) could be pericentromeric (p), interstitial (i), or distal (d). B = banded, DB = dotted-banded, NS = no signal, Chrom = number of chromosomes displaying bands for a given TR family. CPP = *Chorthippus parallelus parallelus*.

	CPP Chromosomes										
TR_Name	Pattern	1	2	3	4	5	6	7	8	X	Chrom
CpaTR004-335	B	d	d	d			d		d		5
CpaTR006-11	B	id	id	id	id	d	d	d	d *	id	9
CpaTR007-21	B	id	d	d	id	d	d	d	d	d	9
CpaTR008-331	B						d *	d *			2
CpaTR009-172	B							i			1
CpaTR010-275	DB							i			1
CpaTR011-213	B		p	p *	p	p	p		p	p	7
CpaTR012-247	B	d	d	d	d	d	d	d	d		8
CpaTR013-55	B				i						1
CpaTR016-7	B					d	d	d	d		4
CpaTR017-289	B		p	d						d	3
CpaTR020-246	B									pi	1
CpaTR024-210	B						pi	p *	p *		3
CpaTR025-248	B						d *	d *			2
CpaTR026-239	B	i					i				2
CpaTR028-148	B		p						p		2
CpaTR030-79	B	p	p *	p							3
CpaTR032-20	B		d	d						d	3
CpaTR033-222	NS										0
CpaTR034-61	NS										0
CpaTR036-168	B	p	p	p	p	p	p	p *	p		8
CpaTR039-139	B		iii	i							2
CpaTR040-161	B	p *i *d *	i	i	p *i *d *	i *	i				6
CpaTR044-205	B	i									1
CpaTR046-157	B	p		p *					d *		3
CpaTR047-287	NS										0
CpaTR048-15	B						id				1
CpaTR049-215	B			i							1
CpaTR050-288	NS										0
CpaTR053-405	B						i				1
CpaTR057-102	NS										0
CpaTR058-196	B									i	1
CpaTR061-27	B									i	1
CpaTR062-56	B									p	1
CpaTR063-92	B								p		1
CpaTR065-379	B						d				1
CpaTR068-155	B	i					i				2
CpaTR069-89	B						i				1
CpaTR074-186	B						d				1
CpaTR075-45	B	d *									1
CpaTR076-168	NS										0
CpaTR077-16	B						id				1
CpaTR091-49	B		d								1
CpaTR094-170	NS										0
CpaTR097-108	NS										0
CpaTR100-295	NS										0
CpaTR103-33	NS										0
CpaTR104-269	B						d				1
CpaTR107-237	NS										0
CpaTR110-159	B							p *			1

**Table 3 genes-14-00397-t003:** Chromosome location of 50 TR families in the CPE subspecies of *C. parallelus*. Asterisks (*) indicate that bands were observed on only one member of a chromosomal pair. Chromosome location (pattern) could be pericentromeric (p), interstitial (i), or distal (d). B = banded, DB = dotted-banded, NS = no signal, Chrom = number of chromosomes displaying bands for a given TR family. CPE = *Chorthippus parallelus erythropus*.

	CPE Chromosomes										
TR_Name	Pattern	1	2	3	4	5	6	7	8	X	Chrom
CpaTR004-335	B	d	d	d							3
CpaTR006-11	B	d		id	id	d	d	d	d	i	8
CpaTR007-21	B	d	d *	d	id	d	d	d	d	id	9
CpaTR008-331	B							pd *			1
CpaTR009-172	B							i			1
CpaTR010-275	DB							i			1
CpaTR011-213	B	p	p	p	p	p	p	p	p	p	9
CpaTR012-247	B	d	d	d	d	d	d	d	d	ii	9
CpaTR013-55	NS										0
CpaTR016-7	B					d	d	d			3
CpaTR017-289	B		p	d							2
CpaTR020-246	B									p	1
CpaTR024-210	B						pi	p			2
CpaTR025-248	NS										0
CpaTR026-239	B	i					i	i *			3
CpaTR028-148	B		p						p		2
CpaTR030-79	B	p	p	p							3
CpaTR032-20	B			d							1
CpaTR033-222	B	d									1
CpaTR034-61	B			i							1
CpaTR036-168	B	p	p *	p	p *		i *				5
CpaTR039-139	B		iii	i							2
CpaTR040-161	B	i			p *i *						2
CpaTR044-205	B	i									1
CpaTR046-157	B			p							1
CpaTR047-287	B						p *	p *		i	3
CpaTR048-15	B						id				1
CpaTR049-215	B			i							1
CpaTR050-288	B	p									1
CpaTR053-405	NS										0
CpaTR057-102	B							p			1
CpaTR058-196	NS										0
CpaTR061-27	B									i	1
CpaTR062-56	B									p	1
CpaTR063-92	B								p		1
CpaTR065-379	B						d				1
CpaTR068-155	B	i *					i				2
CpaTR069-89	B						i				1
CpaTR074-186	B						d				1
CpaTR075-45	B	d	d								2
CpaTR076-168	B							d			1
CpaTR077-16	B						id	i *			2
CpaTR091-49	B		d								1
CpaTR094-170	B			i							1
CpaTR097-108	B	i									1
CpaTR100-295	B							p			1
CpaTR103-33	B	d *									1
CpaTR104-269	B		ii *				id *	i *		i	4
CpaTR107-237	B							p			1
CpaTR110-159	B							p			1

**Table 4 genes-14-00397-t004:** Genotypic and allelic frequencies observed in the hybrid zone (HZ) transect analysed. N = non-banded chromosomes, B = banded chromosomes.

	CpaTR100-295 (Autosomal)								CpaTR104-269 (Sex-Linked)				
	Genotypes				Chromosomes		Allele Frequency		X Chromosomes			Allele Frequency	
Population	NN	NB	BB	Total	N	B	p (N)	q (B)	N	B	Total	p (N)	q (B)
ARU (Arudy)	13	0	0	13	26	0	1	0	8	0	8	1	0
GAB (Gabas)	7	0	0	7	14	0	1	0	8	0	8	1	0
HER (L’Hermine)	6	1	0	7	13	1	0.93	0.07	8	0	8	1	0
POR (Portalet)	10	1	1	12	21	3	0.88	0.13	9	0	9	1	0
CM (Corral de Mulas)	4	3	1	8	11	5	0.69	0.31	6	4	10	0.60	0.40
PAZ (Camino Pazino)	0	0	10	10	0	20	0	1	0	9	9	0	1
ESC (Escarrilla)	0	0	14	14	0	28	0	1	2	7	9	0.22	0.78

## Data Availability

Consensus sequences for all TR families are deposited in GenBank with accession numbers OQ129773—OQ129924. All data are contained within the article or Appendix A.

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
