# Peer review of "Tandem Repeat DNA Provides Many Cytological Markers for Hybrid Zone Analysis in Two Subspecies of the Grasshopper Chorthippus parallelus"

_genes, 2023, doi:10.3390/genes14020397_

Round 1

Reviewer 1 Report

This paper is devoted to how tandem repeat DNA can be useful for the study of hybrid zones (in the case of two subspecies of grasshoppers from Spain and France, which remain in dynamic equilibrium) by specific cytological markers. The authors revealed 50 TR families that could serve as markers for analysis of this HZ, using FISH. Using that approach, they found that the clinal center does not correspond to the geographical center. There are many interesting points in the presented work, the listing of which will take a lot of time 

I am rely on the fact, that the article should be interesting to a wide range of readers.  

In Introduction. Just a recommendation. I would add what is a Library. What do you mean by this? Only one line.

line 44. “Common ancestor 23 Mya” – in [4] - 22.8 Mya- May be about 23 Mya.  

line 80. Cline analysis - It is probably necessary to give a reference to the work that best characterizes this approach. to make the reader immediately understand what you are talking about.    

 line 124. Can you also provide the date of the database used? For today the database is already “RepBase27.12.fasta". Your article may be interesting for a long time 

line 156. “RUL” - here's a repeat of the abbreviation transcription 

line 158. “...  equal to 7 and 6, respectively ... ” - bp?  

Fig. 8. on “c” - PORT. -POR?  

line 294. - PORT.  

lines 350-354. Probably, it should be clarified, that this can also be a bottleneck effect.

Author Response

Reviewer 1

This paper is devoted to how tandem repeat DNA can be useful for the study of hybrid zones (in the case of two subspecies of grasshoppers from Spain and France, which remain in dynamic equilibrium) by specific cytological markers. The authors revealed 50 TR families that could serve as markers for analysis of this HZ, using FISH. Using that approach, they found that the clinal center does not correspond to the geographical center. There are many interesting points in the presented work, the listing of which will take a lot of time.

I am rely on the fact, that the article should be interesting to a wide range of readers.

In Introduction. Just a recommendation. I would add what is a Library. What do you mean by this? Only one line.

AUTHORS’ RESPONSE: We have explained the library hypothesis and added the Fry and Salser (1977) reference which was missing. Thanks for this comment! Therefore, we have modified the beginning of the third paragraph in the Introduction like this:

“The library hypothesis was proposed by Fry and Salser (1977) for satellite DNA, but is applicable to any type of non-coding tandem repeat (TR). It suggests that species within a given taxonomic group (e.g. acridid grasshoppers) share a common library of satellite DNA sequences (or any other TR) whose evolutionary changes are mostly quantitative. On this basis, it predicts that TRs should show low divergence between individuals, …”

line 44. “Common ancestor 23 Mya” – in [4] - 22.8 Mya. - May be about 23 Mya.

AUTHORS’ RESPONSE: Done by adding “about” before 23 Mya.

line 80. Cline analysis - It is probably necessary to give a reference to the work that best characterizes this approach. to make the reader immediately understand what you are talking about.

AUTHORS’ RESPONSE: Done – we added two references, see [23,24]. Thanks.

line 124. Can you also provide the date of the database used? For today the database is already “RepBase27.12.fasta". Your article may be interesting for a long time.

AUTHORS’ RESPONSE: Thanks for this comment. We added “(last accessed 04/27/2022)” in the Methods section

line 156. “RUL” - here's a repeat of the abbreviation transcription.

AUTHORS’ RESPONSE: Done! Thanks for this warning.

line 158. “... equal to 7 and 6, respectively ... ” - bp?

AUTHORS’ RESPONSE: Done! Thanks for this warning.

Fig. 8. on “c” - PORT. -POR?

AUTHORS’ RESPONSE: Corrected to POR. Thanks again!

line 294. - PORT.

AUTHORS’ RESPONSE: Done. Now it says “POR”

lines 350-354. Probably, it should be clarified, that this can also be a bottleneck effect.

AUTHORS’ RESPONSE: It was difficult to localize line numbers mentioned by the reviewer, as they did not coincide with those in our PDF file. Therefore, it is difficult to know to which part of the manuscript refers he/she which could also be a bottleneck effect. Anyway, all contact zones are secondary and respond to climate changes making inhospitable the places sited at higher altitude, so that populations at these sites necessarily pass through a bottleneck. When the climate is again hospitable, the sites are recolonized again and the contact zone is re-established.

Reviewer 2 Report

In this MS, the authors detected a few TR sequences in two subspecies of grasshoppers and studied the HZ of those TRs by using in situ analysis. In my opinion, I think this MS is very readable so I suggest giving it a pass. But several points also need to be addressed.

1) I suggest the authors give a brief introduction/discussion on the TRs and their HZ ability especially their applications.

2) The authors should mention the collection time and the raising methods for those insects (if applicable).

3) Scales are missing.

Author Response

Reviewer 2

In this MS, the authors detected a few TR sequences in two subspecies of grasshoppers and studied the HZ of those TRs by using in situ analysis. In my opinion, I think this MS is very readable so I suggest giving it a pass. But several points also need to be addressed.

1) I suggest the authors give a brief introduction/discussion on the TRs and their HZ ability especially their applications.

AUTHORS’ RESPONSE: We thank the reviewer for this suggestion, and we have added the following sentences to the beginning of the last paragraph in the discussion section:

“Our present analysis has shown that TR markers can be useful for HZ analysis. As the main intra- and inter-population variation for TRs is quantitative, as predicted by the library hypothesis (Fry and Salser 1977), some TRs can display differential amplification in different populations, thus providing markers in the form of FISH bands being very useful to monitorize the evolution of diverging populations or subspecies. In the case of C. parallelus, further studies involving other locations...”

2) The authors should mention the collection time and the raising methods for those insects (if applicable).

AUTHORS’ RESPONSE: Collection time was summer 2016, and the insects were not raised. We mention that now. Thanks for the suggestion.

3) Scales are missing.

AUTHORS’ RESPONSE: Done! Thanks for this warning.